

# High-resolution optical constants of crystalline ammonium nitrate for infrared remote sensing of the Asian Tropopause Aerosol Layer

Robert Wagner[1], Baptiste Testa[2], Michael Höpfner[1], Alexei Kiselev[1], Ottmar Möhler[1], Harald Saathoff[1], Jörn Ungermann[3], and Thomas Leisner[1]

[1]Institute of Meteorology and Climate Research, Karlsruhe Institute of Technology, Karlsruhe, Germany
[2]Department of Physics and Chemistry, University of Lyon, Lyon, France
[3]Institute of Energy and Climate Research, Stratosphere, Forschungszentrum Jülich, Jülich, Germany

*Correspondence to*: Robert Wagner (robert.wagner2@kit.edu)

**Abstract.** Infrared spectroscopic observations have shown that crystalline ammonium nitrate (AN) particles are an abundant constituent of the upper tropospheric aerosol layer which is formed during the Asian summer monsoon period, the so-called Asian Tropopause Aerosol Layer (ATAL). At upper tropospheric temperatures, the thermodynamically stable phase of AN is different from that at 298 K, meaning that presently available room-temperature optical constants of AN, that is, the real and imaginary parts of the complex refractive index, cannot be applied for the quantitative analysis of these infrared measurements. In this work, we have retrieved the first low-temperature data set of optical constants for crystalline AN in the 800 – 6000 cm$^{-1}$ wavenumber range with a spectral resolution of 0.5 cm$^{-1}$. The optical constants were iteratively derived from an infrared extinction spectrum of 1 μm-sized AN particles suspended in a cloud chamber at 223 K. The uncertainties of the new data set were carefully assessed in a comprehensive sensitivity analysis. We show that our data accurately fit aircraft-borne infrared measurements of ammonium nitrate particles in the ATAL.

## 1 Introduction

The term "Asian Tropopause Aerosol Layer" (ATAL) has been established in 2011 when analyses of CALIPSO lidar measurements have revealed the existence of a layer of enhanced aerosol concentrations at altitudes from 13 to 18 km during the Asian summer monsoon (Vernier et al., 2011). The formation of the ATAL is related to the strong convection within the Asian monsoon system, leading to the transport of boundary layer pollutants to the tropical tropopause layer (Vernier et al., 2018). Since its original discovery, the ATAL has been investigated by a variety of balloon-borne, aircraft-borne, and long-term satellite measurements (Höpfner et al., 2019; Lau et al., 2018; Thomason and Vernier, 2013; Vernier et al., 2018; Vernier et al., 2015; Yu et al., 2017). Additionally, chemical transport models have been used to simulate the chemical composition and concentration of aerosol particles lifted to and formed in the ATAL (Fairlie et al., 2020; Gu et al., 2016).



The model simulations predicted that nitrate aerosol is a prominent constituent of the ATAL. Ion chromatographic analyses of filter samples collected during the BATAL campaigns (Balloon Measurements of the ATAL) (Vernier et al., 2018), as well as in situ aerosol mass spectrometer measurements during high-altitude research aircraft flights within the StratoClim project (Stratospheric and upper tropospheric processes for better climate predictions) (Höpfner et al., 2019), confirmed the presence of nitrate aerosol particles. Another important step forward in understanding the chemical composition of the ATAL was made

by the analysis of infrared limb observations from various satellite missions dating back as far as 1997 and from the above-mentioned StratoClim research flights over Nepal and India in 2017 (Höpfner et al., 2019). A distinct infrared signature at 831 $cm^{-1}$ with a width of about 3 $cm^{-1}$, consistently observed for spectra recorded in the ATAL during the Asian summer monsoon, was ascribed to the $\nu_2(NO_3^-)$ mode of crystalline ammonium nitrate (AN), suggesting that AN is an ubiquitous part of the ATAL. The presence of $NH_3$ in the upper troposphere during the Asian summer monsoon, as required for the neutralization of

nitric acid and the formation of AN particles (Wang et al., 2020), had already been detected by previous satellite observations (Höpfner et al., 2016).

        To support the interpretation of the satellite and aircraft-based infrared measurements, laboratory infrared extinction spectra of crystalline AN particles and supercooled aqueous AN solution droplets have been recorded at temperature conditions of the

upper troposphere in the aerosol and cloud simulation chamber AIDA (Aerosol Interaction and Dynamics in the Atmosphere) (Höpfner et al., 2019). These measurements confirmed that the signature at 831 $cm^{-1}$ is due to AN particles in the crystalline state, because the respective $\nu_2(NO_3^-)$ mode of supercooled aqueous AN solution droplets is shifted to 829 $cm^{-1}$ and has a larger width of about 6 $cm^{-1}$ (Höpfner et al., 2019). The crystallization of AN from aqueous AN solution droplets can be induced by various heterogeneous mechanisms, including contact efflorescence (Davis et al., 2015), inclusion of

insoluble dust grains (Han et al., 2002), or admixing small amounts of ammonium sulfate (AS), which precipitates first and triggers the crystallization of AN (Schlenker et al., 2004; Schlenker and Martin, 2005). The phase state of AN is crucial for predicting the particles' impact on cloud formation in the upper troposphere. Whereas supercooled aqueous AN solution droplets only favor cirrus formation by homogeneous ice nucleation (Cziczo and Abbatt, 2001; Koop et al., 2000), crystalline AN particles induce heterogeneous ice formation at much lower relative humidities compared to homogeneous freezing

conditions (Shilling et al., 2006; Wagner et al., 2020).

        To enable the quantitative analysis of the infrared limb observations with respect to the total mass of AN, the infrared $\nu_2(NO_3^-)$ mode recorded in the AIDA chamber has been scaled by the particles' mass concentration to yield mass-specific absorption coefficients (Höpfner et al., 2019). More versatile input parameters for the accurate retrieval of aerosol parameters

from infrared remote sensing measurements are the so-called optical constants, that is, the real and imaginary parts of the complex refractive index (Bohren and Huffman, 1983; Zasetsky et al., 2007). To date, there exists only one data set of the complex refractive index of solid ammonium nitrate at infrared wavelengths (Jarzembski et al., 2003). The data were derived from the analysis of room-temperature infrared spectra of AN compiled in a spectroscopic library, measured using powders



dispersed in an inert paraffin oil. For two reasons, however, this data set cannot be applied for the analysis of the infrared limb
observations of the ATAL. First, the optical constants are not provided at sufficient spectral resolution to mimic the relatively
narrow $\nu_2(NO_3^-)$ mode of AN at 831 cm$^{-1}$. Second, crystalline AN is known to undergo various thermal phase transitions
(Chellappa et al., 2012; Herrmann and Engel, 1997). Specifically, there are different thermodynamically stable phases at room
temperature (phase IV) and upper tropospheric temperatures (phase V, stable below 255 K), going along with marked changes
of the habitus of the corresponding infrared spectra in certain wavenumber regimes, e.g. in the region between 1300 and 1500
cm$^{-1}$ with contributions from the $\nu_4(NH_4^+)$ and $\nu_3(NO_3^-)$ modes (Fernandes et al., 1979; Koch et al., 1996; Shen et al., 1993;
Théorêt and Sandorfy, 1964).

In this work, we have derived the first data set of complex refractive indices of crystalline AN at low temperature (223 K) with
a sufficient spectral resolution (0.5 cm$^{-1}$) to resolve the $\nu_2(NO_3^-)$ mode. The optical constants were deduced for the wavenumber
range from 800 to 6000 cm$^{-1}$, so that they can be employed for the analysis of measurements in any appropriate region in the
mid-infrared not too strongly affected by trace gas signatures. The complex refractive indices were iteratively derived from an
infrared extinction spectrum of about 1 μm-sized, almost pure crystalline AN particles suspended in the AIDA chamber at 223
K (Sects. 2 and 3.1). A comprehensive sensitivity analysis shows the effect of measurement uncertainties and different
assumptions on the shape of the AN particles (spherical or aspherical particle habits) on the optical constants deduced (Sect.
3.2). We demonstrate that this new data set is appropriate to accurately fit infrared limb observations of AN in the ATAL (Sect.
3.3).

## 2 Methods

### 2.1 Experimental setup

The experimental setup has been described in detail in our preceding articles (Höpfner et al., 2019; Wagner et al., 2020) and
will only briefly be summarized here. Figure 1 shows a scheme of the AIDA aerosol and cloud chamber facility with the
instrumentation employed in the present study (Möhler et al., 2003; Wagner et al., 2006b; Wagner et al., 2009). The 84 m$^3$
sized aerosol vessel, placed inside a temperature controlled housing, was cooled to a temperature of 223 K to record the low-
temperature infrared extinction spectrum of crystalline AN particles in their thermodynamic phase V. The mean gas
temperature was recorded by averaging the measurements of four vertically oriented thermocouple sensors mounted at
different heights of the chamber with an uncertainty of ± 0.3 K. The RH$_w$ (relative humidity with respect to supercooled water)
was controlled to 22% by evaporating an equivalent amount of purified water (GenPure Pro UV ultrapure water system,
Thermo Scientific) into the chamber. The resulting water vapor partial pressure, as measured by scanning the intensity of the
rotational-vibrational water vapor absorption line at 1.37 μm with a tunable diode laser (TDL) (Fahey et al., 2014), was divided
by the saturation water vapor pressure over liquid supercooled water to compute RH$_w$ (Murphy and Koop, 2005).





After the AIDA chamber was set to the desired temperature and relative humidity conditions, the injection of the mixed AN/AS aerosol particles was started. As described above, a small admixture of AS was needed to catalyze the crystallization of AN. We prepared an aqueous solution of 99.4 mol% AN and 0.6 mol% AS with an overall solute concentration of about 9 g/100 ml by dissolving respective amounts of AN (99%, VWR Chemicals) and AS (99.5% Merck) in purified water. The aqueous solution was aerosolized with an ultrasonic nebulizer (GA 2400, SinapTec). The particle flow of the mixed AN/AS solution
droplets was first dried to $RH_w \leq 3\%$ with a set of silica gel diffusion dryers (Topas GmbH) and then injected into the cooled AIDA chamber.

In our preceding studies, we have analyzed in detail the composition-dependent crystallization behavior of the mixed AN/AS solution droplets by varying the amount of AS and have found three different scenarios (Höpfner et al., 2019; Wagner et al.,
2020). (i) Pure, 100 mol% AN solution droplets did not crystallize at all and could be maintained for at least 4 hr in the supercooled liquid state at 223 K. (ii) Mixed AN/AS solution droplets with an AS admixture of $\geq 10$ mol% already crystallized during the short transit time through the diffusion dryers and were immediately present as crystalline particles upon injection into the AIDA chamber. (iii) Mixed AN/AS solution droplets with smaller AS admixtures of 0.6 and 2.9 mol% gradually crystallized at low temperature inside the AIDA chamber, with the crystallization rate depending on $RH_w$. We used two in situ
techniques to monitor the ongoing crystallization of the AN/AS solution droplets in the AIDA chamber, namely infrared extinction spectroscopy as well as laser light scattering and depolarization measurements. A Fourier Transform Infrared (FTIR) spectrometer (IFS66v, Bruker) was coupled to an internal multiple reflection cell to measure aerosol infrared extinction spectra over an optical path length of 166.8 m at wavenumbers between 800 and 6000 cm$^{-1}$ with a resolution of 0.5 cm$^{-1}$ (Wagner et al., 2006a). The gradual crystallization of the AN/AS solution droplets was indicated by the continuous decrease of the liquid
water extinction band at about 3500 cm$^{-1}$. The near backscattering linear depolarization ratio, $\delta$, of the aerosol particles at a scattering angle of 178° and a wavelength of 488 nm was measured by the light scattering instrument SIMONE (Schnaiter et al., 2012). A continuous increase of $\delta$ over time was indicative of the formation of aspherical, crystalline AN particles and the concomitant loss of spherical, non-depolarizing solution droplets. The crystallization of the aerosol population was completed when a steady-state $\delta$ value was reached.


With regard to the crystallization experiment with the 99.4 mol% AN and 0.6 mol% AS solution droplets, the time series of $\delta$ indicated that the entire particle population had crystallized within a period of 1 hr after aerosol injection (see Fig. 2a in Wagner et al. (2020)). The infrared extinction spectrum of the almost pure, crystallized AN particles is shown in Fig. 2a of our article, including the AN vibrational assignment according to Fernandes et al. (1979). The minor fraction of AS likely crystallizes as
one of the double salts 2AN·AS or 3AN·AS (Bothe and Beyer, 2007; Schlenker and Martin, 2005), but their specific extinction signatures, e.g. the additional sulfate mode at about 1100 cm$^{-1}$ (Wagner et al., 2020), are barely visible. We can therefore treat the infrared measurement to a good approximation as a pure AN spectrum.



Concomitantly with their infrared spectrum, we measured the particles' number size distribution by combining the size spectra
from a scanning mobility particle sizer (SMPS, model 3071A, TSI, mobility diameter range 0.014 – 0.82 µm) and an
aerodynamic particle spectrometer (APS, model 3321, TSI, aerodynamic diameter range 0.523 – 19.81 µm). To convert the
mobility diameter of the crystalline AN particles from the SMPS measurement into the equal-volume sphere diameter, $d_p$, we
adopted a dynamic shape factor, $\chi$, of 1.1 to account for the slightly aspherical particle habits of the AN crystals (Hinds, 1999).
The same value for $\chi$ as well as the particle density of AN, $\rho(AN) = 1.72\ \mathrm{g\ cm^{-3}}$, were chosen to transform the aerodynamic
diameter from the APS measurement into $d_p$ (Hinds, 1999; Kelly and McMurry, 1992). The resulting number size distribution
as a function of $d_p$ is shown in Fig. 2b. The dominant particle mode mostly falls into the measurement range of the APS
instrument and is centered at $d_p = 0.98$ µm. The integrated total number concentration of the AN particles, $N_{aer}$, is 985 $\mathrm{cm^{-3}}$, in
good agreement with an independent measurement of $N_{aer}$ with a condensation particle counter (CPC, model 3010, TSI). The
particle size measurement from Fig. 2b and the infrared spectrum from Fig. 2a are the basic input parameters for the retrieval
scheme of the optical constants (Sect. 2.2).

Another important parameter for the implementation of the retrieval approach is the shape of the crystalline AN particles. We
have therefore recorded electron microscope images of filter-collected AN particles from the AIDA chamber (Fig. 2c). Note
that these images were obtained from a different experiment with crystallized particles from 97.1 mol% AN and 2.9 mol% AS,
but we do not expect that the slightly higher AS content significantly alters the particle shape. For particle sampling from the
AIDA chamber, we have developed a method to collect aerosol particles in the cryo-preserved state (Wagner et al., 2017). For
that purpose, we used a vacuum cryo-transfer system (EM VCT500, Leica) to ensure an unbroken chain of cryogenic transfer
steps from particle sampling to electron microscopic analysis (Fig. 1). This avoids the polymorphic phase change of the solid
AN particles from phase V to IV, which might induce a change of particle morphology. The cryo-transfer unit includes a
liquid-nitrogen cooled transfer shuttle with a silicon substrate mounted on a copper sample holder. For particle sampling, the
Si substrate was introduced through a pneumatic air lock into a sampling chamber located inside the isolating containment of
the AIDA chamber. The sampling of the AN particles was electrostatically assisted by directing the aerosol flow from the
AIDA chamber through a Polonium neutralizer (model 3077A, TSI) onto the sampling substrate connected to a high voltage
power supply set to 2 kV. After sampling, the substrate was retracted into the pre-cooled shuttle without exposing it to the
ambient air. Thereafter, the shuttle was detached from the AIDA air lock and transferred under continuous supply of liquid
nitrogen to the ESEM (environmental scanning electron microscope) laboratory, where it was introduced through the air lock
into the microscope (Quattro S, FEI ThermoFisher Scientific). The sample imaging was conducted in the $N_2$ atmosphere, thus
avoiding sputtering of the particle surface normally required for high-resolution imaging in a SEM. The electron microscope
images shown in Fig. 2c reveal that the crystalline AN particles are of rather compact shape with aspect ratios predominantly
in the range from 0.80 to 1.25. Similar morphologies have been reported for crystallized AS particles (Earle et al., 2006).





## 2.2 Retrieval scheme for deriving the optical constants

The retrieval scheme, as sketched in Fig. 3, follows standard procedures for the derivation of optical constants from aerosol infrared extinction spectra (Dohm and Niedziela, 2004; Earle et al., 2006; Norman et al., 1999; Segal-Rosenheimer et al., 2009; Signorell and Luckhaus, 2002; Zasetsky et al., 2005). Briefly, an initial guess for the wavenumber-dependent spectrum of the
imaginary part of the complex refractive index, $k(\tilde{v})$, was derived by subtracting the scattering contribution from the measured infrared extinction spectrum (step 1). Using the subtractive Kramers-Kronig transformation (Ahrenkiel, 1971; Milham et al., 1981), we then computed the real part of the complex refractive index, $n(\tilde{v})$, for each wavenumber grid point of the measured spectrum (step 2). In its subtractive form, the Kramers-Kronig relation needs a so-called anchor point, a known value for the real index of refraction at some reference wavenumber, $n(\tilde{v}_x)$. As a reasonable estimate for the anchor point, we employed a
value of $n = 1.56$ at 4600 cm$^{-1}$ from the tabulated room-temperature refractive indices by Jarzembski et al. (2003). The anchor point value was included as one of three parameters in our sensitivity analysis to investigate the variability of the retrieval results for the optical constants on the uncertainty ranges of the input parameters (Sect. 3.2). In this analysis, we also considered two other input values for $n(\tilde{v}_x)$, namely 1.52 and 1.60.

In step 3 of the retrieval scheme, the $n(\tilde{v})$ and $k(\tilde{v})$ spectra as well as the size distribution of the AN particles were fed into the optical model to compute the extinction spectrum. The size distribution measurement involved another sensitivity parameter, namely the dynamic shape factor $\chi$ (Hinds, 1999). As explained in Sect. 2.1, this parameter was needed to account for the effect of shape on particle motion when converting the measured mobility and aerodynamic diameters of the crystalline AN particles into equal-volume sphere diameters as input for the optical model (Fig. 2b). Our best estimate for $\chi$ was a value
of 1.1, as representative for slightly aspherical particle habits (Hinds, 1999). For $\chi = 1.1$, we obtained a median particle diameter of 0.98 μm and a total aerosol volume concentration, $V_{aer}$, of 410 μm$^3$ cm$^{-3}$. In our sensitivity analysis (Sect. 3.2), we have repeated the retrieval procedure of the optical constants with two other $\chi$ values, namely 1.05 and 1.15, in order to test the impact of the uncertainty of $\chi$ on the deduced refractive indices. For $\chi = 1.05$, the median diameter of the dominant particle mode is shifted to 0.96 μm, decreasing $V_{aer}$ by about 7% to 380 μm$^3$ cm$^{-3}$, whereas for $\chi = 1.15$, it is shifted to 1.00 μm,
increasing $V_{aer}$ by about 7% to 440 μm$^3$ cm$^{-3}$.

The third sensitivity parameter was the particle aspect ratio, $\phi$, employed in the optical model. We first derived the optical constants using Mie theory under the assumption of spherical particles ($\phi = 1$). In the sensitivity analysis, we additionally used the T-matrix code to model the AN particles as randomly-oriented spheroids with six different aspect ratios between $\phi = 0.5$
and 2.0 (Mishchenko and Travis, 1998). After the computation of the extinction spectrum (step 4), we calculated the root-mean square deviation (RMSD) between calculated and measured infrared spectrum (step 5), and finished the loop by iteratively adjusting the initial guess $k(\tilde{v})$ spectrum to minimize the RMSD (step 6). The computational details of the retrieval



scheme are described in Appendix A. Figure A1 therein shows the good agreement between calculated and measured infrared spectrum after the minimization procedure.

## 3 Results and discussion

### 3.1 The new low-temperature data set of optical constants for crystalline AN

Figure 4 shows our newly derived, low-temperature complex refractive index data set for crystalline AN, as obtained when employing the parameter values $n(\tilde{\nu}_x) = 1.56$, $\chi = 1.1$, and $\phi = 1$ in the retrieval scheme (black line). In comparison with the room-temperature data from Jarzembski et al. (2003) (red line), the general magnitude of the imaginary indices in the various absorption regimes is similar, with maximum $k$ values of about 0.5, 2, and 1 in the wavenumber regimes 2800 – 3500 cm$^{-1}$, 1250 – 1500 cm$^{-1}$, and 820 – 840 cm$^{-1}$, respectively. The low-temperature $k(\tilde{\nu})$ spectrum reveals more fine structure in the 2800 – 3500 cm$^{-1}$ and 1250 – 1500 cm$^{-1}$ absorption regions compared to the room-temperature data set. Very similar temperature-dependent spectral changes have been observed between the infrared spectra of AN films recorded at 270 and 90 K (Koch et al. (1996), see spectra B and C in their Fig. 4), and were attributed to the temperature-induced polymorphic phase change of AN between phase IV and phase V. The peak positions of the infrared absorption bands in our low-temperature $k(\tilde{\nu})$ spectrum, that is, 3235 cm$^{-1}$ for $\nu_3(NH_4^+)$, 3062 cm$^{-1}$ for $(\nu_2+\nu_4)(NH_4^+)$, 1760 cm$^{-1}$ for $\nu_2(NH_4^+)$, 1055 cm$^{-1}$ for $\nu_1(NO_3^-)$, and 831 cm$^{-1}$ for $\nu_2(NO_3^-)$, are in good agreement with the tabulated values for ammonium nitrate films probed at 90 K (Koch et al., 1996). In contrast, we observed a different spectral pattern in the 1250 – 1500 cm$^{-1}$ regime, with band maxima centered at 1356, 1418, and 1476 cm$^{-1}$, whereas peak positions at 1320, 1367, 1397, 1462, and 1492 cm$^{-1}$ were reported by Koch et al. (1996). They emphasized that this particular wavenumber region was very sensitive to the procedure by which the AN films were deposited, arguing that the $\nu_4(NH_4^+)$ and $\nu_3(NO_3^-)$ modes are prone to site-sensitive coupling due to their similar frequencies. Regarding our $k(\tilde{\nu})$ spectrum, we will show in Sect. 3.2 that the 1250 – 1500 cm$^{-1}$ wavenumber regime is also particularly sensitive to the variation of the parameter values $n(\tilde{\nu}_x)$, $\chi$, and $\phi$, which contributes to the observed frequency and intensity shifts with respect to the infrared spectrum of the AN film in that region. With regard to the $\nu_2(NO_3^-)$ mode of AN, Koch et al. (1996) did not observe any frequency shift upon cooling from 270 to 90 K. Also, room-temperature infrared measurements of solid AN particles pointed to the same band position at 831 cm$^{-1}$ as in our low-temperature spectrum (Schlenker and Martin, 2005). The apparent shift of this band in the $k$ spectrum reported by Jarzembski et al. (2003) (Fig. 4c) might therefore be related to the insufficient spectral resolution of the underlying infrared spectrum used for the retrieval of the optical constants.

### 3.2 Sensitivity analysis

An overview about our sensitivity analysis regarding the effect of different values for the parameters $n(\tilde{\nu}_x)$, $\chi$, and $\phi$ on the retrieved $k(\tilde{\nu})$ spectrum of the AN particles is shown in Fig. 5. In the first part (panels a – c), we varied $n(\tilde{\nu}_x)$ between 1.52





and 1.60 while keeping $\chi$ and $\phi$ constant at 1.1 and 1, respectively. The associated changes in $k(\tilde{\nu})$ were small; most notably, an increasing value for the real refractive index at the anchor point of 4600 cm$^{-1}$ led to slightly decreasing intensities of the

$k(\tilde{\nu})$ spectrum in the N – H stretching mode regime between 2800 and 3500 cm$^{-1}$, whereas the $k$ values of the $\nu_2(NO_3^-)$ mode at 831 cm$^{-1}$ almost remained unaffected. Different values for $\chi$ (panels d – f) immediately affected the size distribution and thereby the overall volume concentration of the AN particles, $V_{aer}$, with lower values for $\chi$ giving rise to lower values for $V_{aer}$ (see Sect. 2.2). Within the approximation that particle absorption is primarily governed by the magnitude of the imaginary index $k$, a lower value for $\chi$, that is, $V_{aer}$, must be balanced by higher $k$ values in the retrieved $k(\tilde{\nu})$ spectrum. Such behavior

was indeed observed in the wavenumber regimes of the moderately intense absorption bands of AN between 2800 and 3500 cm$^{-1}$ as well as 810 and 850 cm$^{-1}$. In particular, the $k$ values were regularly increased or decreased over the whole wavenumber region of the absorption bands when $\chi$ was decreased or increased, respectively.

The above approximation breaks down in the regime of the very intense vibrational modes of AN between 1200 and 1500

cm$^{-1}$. With high values for the imaginary index ($k > 1$), going along with high-amplitude anomalous dispersion signals in the $n$ spectrum (Fig. 4), the resonance condition for inducing so-called Fröhlich, or surface modes can be fulfilled, meaning that $n$ approaches zero and $k$ adopts a value of $\sqrt{2}$ (Bohren and Huffman, 1983; Clapp and Miller, 1993; Leisner and Wagner, 2011). In such a case, there will be an enhanced cross section in the absorption spectrum of small spherical particles and the spectral habitus of an absorption band (including e.g. band intensity and peak position) can strongly differ from the $k(\tilde{\nu})$ spectrum.

Regarding our sensitivity analysis, this meant that the variation of $\chi$, that is, $V_{aer}$, did not lead to a regular, predictable change in the retrieved $k(\tilde{\nu})$ spectrum as in the other wavenumber regimes with the less intense absorption bands. The variation of $\chi$ only affected a certain wavenumber region between 1320 and 1370 cm$^{-1}$ in the retrieval result for $k(\tilde{\nu})$ (Fig. 5e). In Appendix B and the associated Figs. B1 and B2, we provide an extended description of the peculiar signature of the AN infrared spectrum in the 1200 – 1500 cm$^{-1}$ regime.


Frequency and intensity of the surface modes are strongly dependent on the particle shape (Bohren and Huffman, 1983; Clapp and Miller, 1993). We therefore obtained pronounced, shape-dependent variations in the retrieval results for the $k(\tilde{\nu})$ spectrum in the 1200 – 1500 cm$^{-1}$ regime when modifying the aspect ratio of the crystalline AN particles (Fig. 5g). For spherical particles with $\phi = 1$, the two most intense maxima in the retrieved $k(\tilde{\nu})$ spectrum were at 1418 and 1356 cm$^{-1}$, with the latter having the

higher intensity. An increasing degree of asphericity inverted the intensities of these two maxima, with the 1418 cm$^{-1}$ band, slightly shifted to higher wavenumbers, becoming the more intense one, and the 1356 cm$^{-1}$ band, also slightly shifted to higher wavenumbers, becoming the less intense one. These shape-dependent absorption signatures induce a further degree of complexity to the infrared spectrum of AN in the 1200 – 1500 cm$^{-1}$ wavenumber regime, in addition to the variability already highlighted by Koch et al. (1996) regarding the thin film AN spectra. In contrast, the shape-dependent variations of the

retrieved $k(\tilde{\nu})$ spectrum in the regime of the $\nu_2(NO_3^-)$ mode were small (Fig. 5h). The maximum value for $k$ at 831 cm$^{-1}$ varied





by at most 6% between the retrieval results for the various aspect ratios, with a trend of decreasing maximum value with increasing degree of asphericity. The shape-related spectral changes in the $2800 - 3500$ cm$^{-1}$ regime of the $k(\tilde{\nu})$ spectrum were insignificant and are therefore not shown.

### 3.3 Application of the new refractive index data set

We consider the refractive index data set derived for $n(\tilde{\nu}_x) = 1.56$, $\chi = 1.1$, and $\phi = 1$ as our currently best estimate for the optical constants of crystalline AN at 223 K (Fig. 4). However, we also provide the data sets retrieved for all sensitivity studies described in Sect. 3.2 and strongly encourage any modelers who apply our data to take into account these uncertainties of $n$ and $k$ in their own spectral analyses. Regarding the influence of particle asphericity investigated by the T-matrix calculations, we have considered aspect ratios partly exceeding the actual particles' eccentricity as revealed by the electron microscope

images (Fig. 2c) to ensure that we derived a valid upper estimate for the uncertainty of $n$ and $k$ related to the shape dependency of the infrared extinction signatures of AN. As each individual $n$ and $k$ data set relies on an idealized representation for the shape of the AN particles (either spherical or spheroidal with a fixed aspect ratio), modelers might also consider using a shape-averaged $n$ and $k$ data set in their analyses (Mishchenko et al., 1997).

We used the new refractive index data set of AN to re-analyze infrared limb observations of the ATAL with the airborne GLORIA (Gimballed Limb Observer for Radiance Imaging of the Atmosphere) instrument during the StratoClim research flight on 31 July 2017 (Fig. 6a). The AN mass concentrations derived with the $n(\tilde{\nu}_x) = 1.56$, $\chi = 1.1$, and $\phi = 1$ data set agree within $0.01 \pm 0.06$ $(3\sigma)$ µg m$^{-3}$ with those previously estimated using the mass-specific absorption coefficients (see Fig. 3b in Höpfner et al., 2019). Maximum deviations of about $\pm 0.04$ µg m$^{-3}$ for the retrieved AN mass concentrations are obtained in

the sensitivity analysis when considering the uncertainties of the $n$ and $k$ values. The measured infrared spectrum of the $\nu_2(NO_3^-)$ mode of the AN particles (blue line in Fig. 6b), showing the difference of mean GLORIA spectra recorded in periods of high and low AN mass concentrations, is accurately reproduced in the spectral fit with the new optical constants (orange line). Apart from the $\nu_2(NO_3^-)$ mode, our new data set of refractive indices also opens the possibility to exploit bands in other spectral regimes for the detection and quantification of AN in remote sensing observations of the atmosphere.

### Appendix A – Computational details of the retrieval scheme


Here, we describe the computational details of the individual steps in the retrieval scheme for deriving the infrared complex refractive indices of AN (Fig. 3). The wavenumber-dependent complex refractive index, $N(\tilde{\nu})$, is given as $N(\tilde{\nu}) = n(\tilde{\nu}) + ik(\tilde{\nu})$, with the real ($n$) and imaginary ($k$) parts called the optical constants. Our retrieval approach takes advantage of the Kramers-Kronig relation (Bohren and Huffman, 1983), by which the real refractive index can be computed at any given

wavenumber $\tilde{\nu}_k$ from the full wavenumber spectrum of $k(\tilde{\nu})$ (Eq. A1).





$$n\left(\tilde{v}_{k}\right)-1=\frac{2}{\pi}P\int_{0}^{\infty}\frac{k\left(\tilde{v}\right)\tilde{v}}{\tilde{v}^{2}-\tilde{v}_{k}^{2}}d\tilde{v} \tag{A1}$$

In step 1 of our retrieval approach, where we derived an initial guess for the optical constants of AN from the measured infrared extinction spectrum, we made use of a different form of the Kramers-Kronig relation, linking the real and imaginary parts of the composite function $f = (N^{2}(\tilde{v}) - 1)/(N^{2}(\tilde{v}) + 2)$ (Rouleau and Martin, 1991).

$$\mathrm{Re}\{f\}\left(\tilde{v}_{k}\right)=\frac{2}{\pi}P\int_{0}^{\infty}\frac{\mathrm{Im}\{f\}\left(\tilde{v}\right)\tilde{v}}{\tilde{v}^{2}-\tilde{v}_{k}^{2}}d\tilde{v} \tag{A2}$$

For the initial estimate of $k(\tilde{v})$, we analyzed the extinction spectrum of the 1 µm-sized AN crystals in the framework of Rayleigh theory (Bohren and Huffman, 1983). The scattering contribution to extinction was subtracted from the measurement by assuming its intensity to be proportional to $\tilde{v}^{4}$ (Norman et al., 1999). In the Rayleigh limit, the so-derived absorption spectrum is directly proportional to the total aerosol volume concentration, $V_{\mathrm{aer}}$, and to the imaginary part of the composite function $f$, $Im\{f\}$ (Ossenkopf et al., 1992). With $V_{\mathrm{aer}}$ given by the SMPS and APS size distribution measurements as discussed in Sect. 2.1, we could thus derive the full wavenumber spectrum of $Im\{f\}$ from the estimated absorption spectrum, perform the Kramers-Kronig integration in Eq. A2 to obtain the spectrum of $Re\{f\}$, and calculate the initial guess for $k(\tilde{v})$ from $Re\{f\}$ and $Im\{f\}$ (Leisner and Wagner, 2011; Segal-Rosenheimer et al., 2009).

The initial guess $k(\tilde{v})$ spectrum had to be iteratively adjusted using Mie theory or the T-matrix method because the generic requirement for the validity of the Rayleigh approximation, that is, the AN particles have to be very small compared to any of the wavelengths (Bohren and Huffman, 1983), was not fulfilled for the entire measurement range in the mid-infrared. The Kramers-Kronig integration in its form of Eq. A2 was therefore only applied once in step 1 of the retrieval scheme. In step 2, that was part of the loop for the iterative adjustment of $k(\tilde{v})$, we employed the direct relation between $n(\tilde{v})$ and $k(\tilde{v})$, implementing Eq. A1 in the version of the subtractive Kramers-Kronig transformation to minimize the effect of truncation errors due to the unknown behavior of $k(\tilde{v})$ beyond the experimentally accessible wavenumber range (Ahrenkiel, 1971; Milham et al., 1981; Segal-Rosenheimer and Linker, 2009).

$$n\left(\tilde{v}_{k}\right)=n\left(\tilde{v}_{x}\right)+\frac{2\left(\tilde{v}_{k}^{2}-\tilde{v}_{x}^{2}\right)}{\pi}P\int_{0}^{\infty}\frac{k\left(\tilde{v}\right)\tilde{v}}{\left(\tilde{v}^{2}-\tilde{v}_{k}^{2}\right)\left(\tilde{v}^{2}-\tilde{v}_{x}^{2}\right)}d\tilde{v} \tag{A3}$$





As described in Sect. 2.2, the anchor point value, $n(\tilde{v}_x)$, was set to 1.56 at 4600 cm$^{-1}$. Two other values for $n(\tilde{v}_x)$, namely 1.52

and 1.60, were adopted in our sensitivity analysis (Sect. 3.2). Prior to performing the Kramers-Kronig integration, we extended the $k(\tilde{v})$ spectrum below the 800 cm$^{-1}$ cutoff of our infrared measurements with the room-temperature $k$ values reported by Jarzembski et al. (2003). The integral was computed with Maclaurin's formula method (Ohta and Ishida, 1988).

For the Mie calculations, modeling the crystalline AN particles as spheres with an aspect ratio, $\phi$, of one, we extended the Mie

code provided by Bohren and Huffman (1983) to average the computed extinction cross sections over the measured number distribution of particle sizes. Due to their computational efficiency, the Mie computations were explicitly included in each iteration, meaning that for each new adjustment of the $k(\tilde{v})$ and $n(\tilde{v})$ spectra, a new Mie calculation of the extinction spectrum was executed. In order to investigate the effect of particle shape on the retrieval results for the optical constants, we also modeled the AN particles as randomly-oriented spheroids, considering both oblate ($\phi > 1$) and prolate ($\phi < 1$) particle shapes.

To compute the extinction cross sections, we used the extended precision T-matrix code for randomly oriented particles by Mishchenko and Travis (1998), choosing six different aspect ratios, namely $\phi = 1/2$, $\phi = 2/3$, $\phi = 4/5$, $\phi = 5/4$, $\phi = 3/2$, and $\phi = 2$. Here, the computational burden would have been too high to perform a complete T-matrix computation of the size-averaged extinction spectrum of the AN particles each time the $k(\tilde{v})$ and $n(\tilde{v})$ spectra were newly adjusted during the optimization procedure. We therefore computed a priori for each aspect ratio an extinction cross section database on a three-dimensional

parameter space, which served as a look-up table in the retrieval procedure. Specifically, the extinction cross sections were computed for

- 27 equal-volume sphere size parameters $x_p$ ($x_p = \pi d_p/\lambda$) between 0.005 and 5 (0.005, 0.01, 0.03, 0.05, 0.1, 0.2 – 4.0 with $\Delta x_p = 0.2$, 4.5, and 5),
- 30 values for the real refractive index $n$ between 0.4 and 3.3 with $\Delta n = 0.1$, and
- 29 values for the imaginary refractive index $k$ between 0.0001 and 2.5 (0.0001, 0.025, 0.05, 0.075, 0.1 – 2.5 with $\Delta k = 0.1$),

summing up to 23490 individual calculations for each aspect ratio. In the iterative loop of the retrieval scheme, the T-matrix

computed extinction spectrum for a new adjustment of $k(\tilde{v})$ and $n(\tilde{v})$ was then simply obtained by spline interpolation from the pre-computed extinction cross section database. We have validated the accuracy of the interpolation scheme with the denoted number of grid points in the ($x_p$, $n$, $k$) space by the comparison between directly computed and interpolated extinction cross sections.

To minimize the root-mean square deviation between measured and calculated extinction spectrum, we used the downhill simplex algorithm as the optimization technique in the iterative adjustment of the $k(\tilde{v})$ spectrum (Press et al., 1992). The



measured spectrum is provided in a digital resolution of about 0.24 cm$^{-1}$, resulting from an approximately doubled size of the original interferogram due to zero-filling with a factor of two (Aroui et al., 2012). Overall, this amounts to 21572 wavenumber grid points for the $k(\tilde{\nu})$ spectrum in the wavenumber range between 800 and 6000 cm$^{-1}$. The $k(\tilde{\nu})$ spectrum of AN features

regimes with vastly varying types of absorption signatures, including, for example, completely non-absorbing regions, regimes of spectrally broad but weak absorption, and regimes with strong absorption bands, either spectrally broad or narrow. In the retrieval scheme, the various spectral regimes were treated differently and not all wavenumber grid points were included as optimization parameters.

- Wavenumber region 800 – 1250 cm$^{-1}$. This regime includes the spectrally narrow $\nu_2(NO_3^-)$ and $\nu_1(NO_3^-)$ modes at 831 and 1055 cm$^{-1}$. Apart from these two pronounced bands, the absorption is very low. We therefore optimized the $k(\tilde{\nu})$ values in the regimes 826 – 836 cm$^{-1}$ and 1050 – 1060 cm$^{-1}$ with the full spectral resolution. All other $k$ values were set to a small, constant background value of 0.002 in order to not transfer the noise from the baseline in the experimental spectrum to the $k(\tilde{\nu})$ spectrum.

- Wavenumber region 1250 – 1500 cm$^{-1}$. This regime comprises the intense, spectrally broad absorption bands due to the $\nu_4(NH_4^+)$ and $\nu_3(NO_3^-)$ modes. All $k(\tilde{\nu})$ values were included in the optimization, but we applied a weak smoothing function prior to the Kramers-Kronig transformation to avoid the occurrence of singular spikes in the $k(\tilde{\nu})$ spectrum (Savitzky-Golay smoothing filter, quadratic polynomial fit with 5 data points in the moving window) (Press et al., 1992).

- Wavenumber region 1500 – 2800 cm$^{-1}$. This regime includes two very weak, but spectrally narrow absorption signatures between about 1750 and 1785 cm$^{-1}$, presumably due to the $\nu_2(NH_4^+)$ and $(\nu_1+\nu_4)(NO_3^-)$ modes (Fernandes et al., 1979; Koch et al., 1996). In that part, we optimized all $k(\tilde{\nu})$ values without any smoothing filter. Outside the 1750 – 1785 cm$^{-1}$ region, the residual absorption is spectrally broad and very small, so that we either used a strong smoothing filter or set the $k$ values to a small, constant background value of 0.002.

- Wavenumber region 2800 – 3500 cm$^{-1}$. This regime comprises the intense N–H stretching modes and was treated in the same way as the 1250 – 1500 cm$^{-1}$ wavenumber region.

- Wavenumber region 3500 – 6000 cm$^{-1}$: In this non-absorbing regime, all $k(\tilde{\nu})$ values were set to zero.

A comparison between the measured (black lines) and the computed (red lines) infrared extinction spectrum of crystalline AN

particles in five different wavenumber regimes after convergence of the optimization algorithm is shown in Fig. A1. The above description specifically relates to the retrievals performed with Mie theory. Regarding the T-matrix computations, the analysis of the extinction cross section data base showed that the wavenumber region above 1500 cm$^{-1}$ was only prone to minor shape-related changes for the considered range of aspect ratios. Therefore, the analysis of the shape-dependent variations in the retrieval results for the optical constants of AN was confined to the 800 – 1500 cm$^{-1}$ regime (see Figs. 5g and 5h).



**Appendix B – Analysis of the infrared extinction signature of the AN particles between 1200 and 1500 cm$^{-1}$**

In order to better understand the signature of the infrared extinction spectrum of the crystalline AN particles in the 1200 – 1500 cm$^{-1}$ wavenumber regime, we show in Fig. B1 the comparison between the measured extinction spectrum (panel a) and the retrieved optical constants for $n(\tilde{\nu}_x) = 1.56$, $\chi = 1.1$, and $\phi = 1$ (panel b). It is obvious that the spectral signature of the measured extinction bands of the AN particles is clearly different from that of the underlying $k(\tilde{\nu})$ spectrum, that is, different

from a bulk absorption spectrum of AN whose intensity would be proportional to $k(\tilde{\nu}) \cdot \tilde{\nu}$ (Bohren and Huffman, 1983). The two most intense maxima in the $k(\tilde{\nu})$ spectrum at 1418 and 1356 cm$^{-1}$ (dashed red lines) are shifted to higher wavenumbers in the measured particle spectrum (1424 and 1373 cm$^{-1}$, dashed blue lines), accompanied by significant changes in their relative intensities. The intensity of the 1424 cm$^{-1}$ band is disproportionately high compared to that of the 1373 cm$^{-1}$ band, considering that the value of the imaginary refractive index is almost identical at both frequencies ($k \sim 1.4$). However, the real refractive

index at 1424 cm$^{-1}$ is much lower than at 1373 cm$^{-1}$, thereby better fulfilling the resonance condition for the manifestation of a Fröhlich, or surface mode, in the particle spectrum ($n \approx 0$ and $k \approx \sqrt{2}$) (Bohren and Huffman, 1983; Clapp and Miller, 1993). The same reasoning explains the frequency shift of the particle extinction bands towards higher wavenumbers compared to the maxima in the $k(\tilde{\nu})$ spectrum. Although this frequency shift leads to a reduction of the magnitude of the imaginary index $k$, the intensity of the particle extinction bands is enhanced due to the simultaneous decrease of the value for the real refractive

index $n$, yielding a higher cross section due to a better match with the resonance condition. For a less intense absorption band like the $\nu_2(NO_3^-)$ mode at 831 cm$^{-1}$, the amplitude of the anomalous dispersion feature in the $n(\tilde{\nu})$ spectrum is reduced and there are no explicit frequencies where the optical constants match the resonance condition. Therefore, the particle extinction band only experiences a minor frequency shift of about 0.4 cm$^{-1}$ compared to the location of the absorption peak in the $k(\tilde{\nu})$ spectrum (Fig. B2).


In our sensitivity analysis (Sect. 3.2), we have investigated the response of the retrieved $k(\tilde{\nu})$ spectrum for the AN particles to a change of the parameter $\chi$, which controls the overall aerosol volume concentration, $V_{aer}$. We have observed that in the 1200 – 1500 cm$^{-1}$ regime, the variation of $\chi$ did not induce a regular increase or decrease of the $k$ values over the whole frequency range (Fig. 5e). Obviously, the peak at 1418 cm$^{-1}$ in the retrieved $k(\tilde{\nu})$ spectrum is much less influenced by a change in $V_{aer}$

than the absorption band at 1356 cm$^{-1}$. As a tentative explanation, one might argue that the particle extinction band at 1424 cm$^{-1}$, resulting from the $k$ maximum at 1418 cm$^{-1}$, is less affected by $V_{aer}$ because its intensity is primarily governed by the strongly enhanced cross sections resulting from the match of the optical constants with the resonance condition. At 1356 cm$^{-1}$, the resonance condition is of much less importance for the band intensity, meaning that the magnitude of the retrieved $k$ values at this frequency are more directly influenced by a change of $\chi$, that is, $V_{aer}$.

## Data availability

Upon manuscript acceptance, we will archive all refractive index data sets derived in this work in the KITopen repository, the central publication platform for KIT (Karlsruhe Institute of Technology) scientists (Open Access, contact: KITopen@bibliothek.kit.edu), and assign them a citable persistent identifier (DOI).

## Author contributions

Conceptualization, methodology, and supervision: RW, MH. Formal analysis: BT, MH, RW. Investigation: RW, BT, AK, MH, HS, JU. Project administration: OM, TL. Software: BT, MH, RW. Visualization: RW, BT, AK, MH. Writing – original draft: RW, MH. Writing – review & editing: all authors.

## Competing interests

The authors declare that they have no conflict of interest.

## Acknowledgments

We gratefully acknowledge the continuous support by all members of the Engineering and Infrastructure group of IMK-AAF, in particular by Olga Dombrowski, Rainer Buschbacher, Tomasz Chudy, Steffen Vogt, and Georg Scheurig. This work has been funded by the Helmholtz-Gemeinschaft Deutscher Forschungszentren as part of the program "Atmosphere and Climate". Additional funding has been received by the French Erasmus+ agency.

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





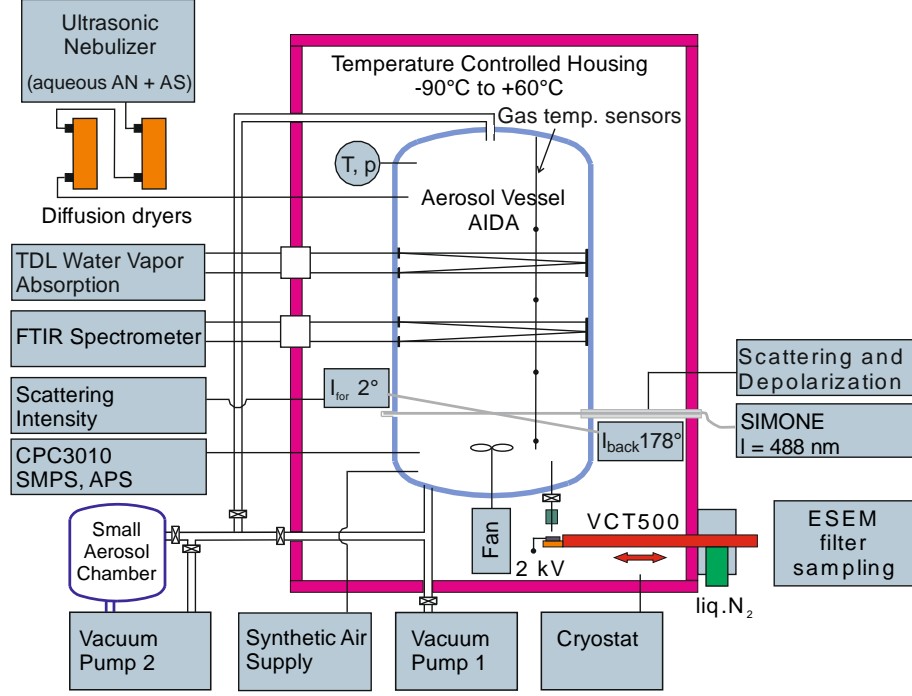


**Figure 1: Scheme of the instrumentation of the AIDA aerosol and cloud chamber facility employed in the present study. The abbreviations are explained in Sect. 2.1.**



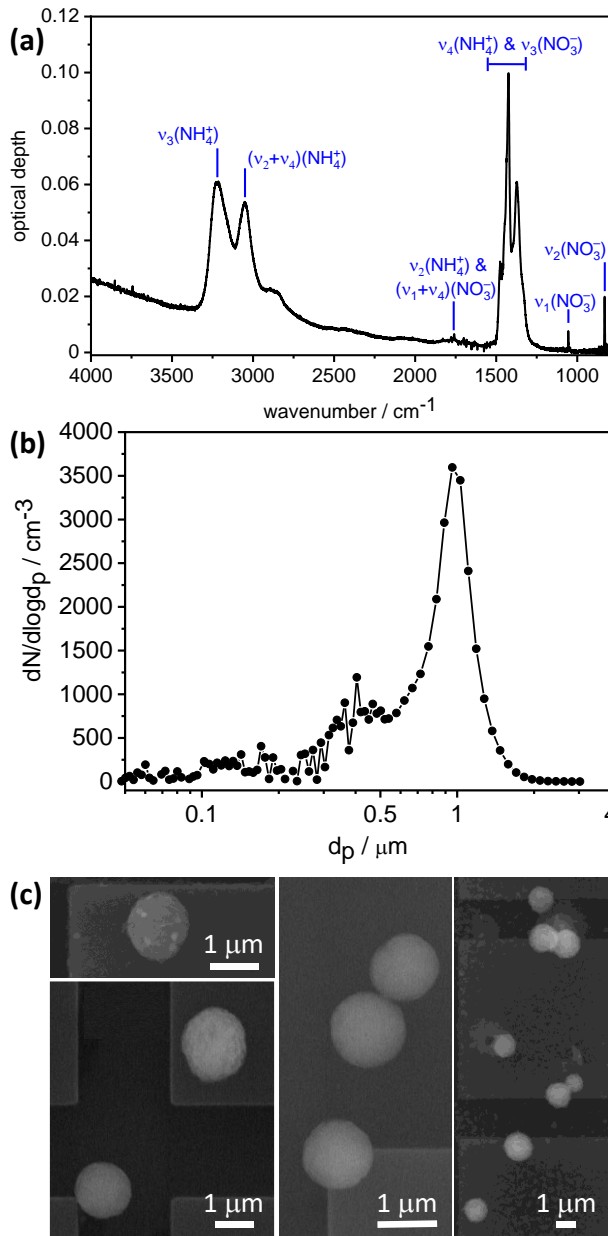

Figure 2: (a) Infrared extinction spectrum of crystalline AN particles recorded at 223 K. The notation of the AN vibrational modes is according to Fernandes et al. (1979). (b) Concomitant number size distribution of the almost pure AN particles crystallized from solution droplets with 99.4 mol% AN and 0.6 mol% AS. The data are shown as a function of the equal-volume sphere diameter, $d_P$, as obtained by converting the mobility and aerodynamic size spectra from the SMPS and APS measurements with $\rho(AN) = 1.72 \text{ g cm}^{-3}$ and $\chi = 1.1$. (c) Exemplary electron microscope images of filter-sampled, continuously cooled AN crystals from a crystallization experiment with 97.1 mol% AN and 2.9 mol% AS.





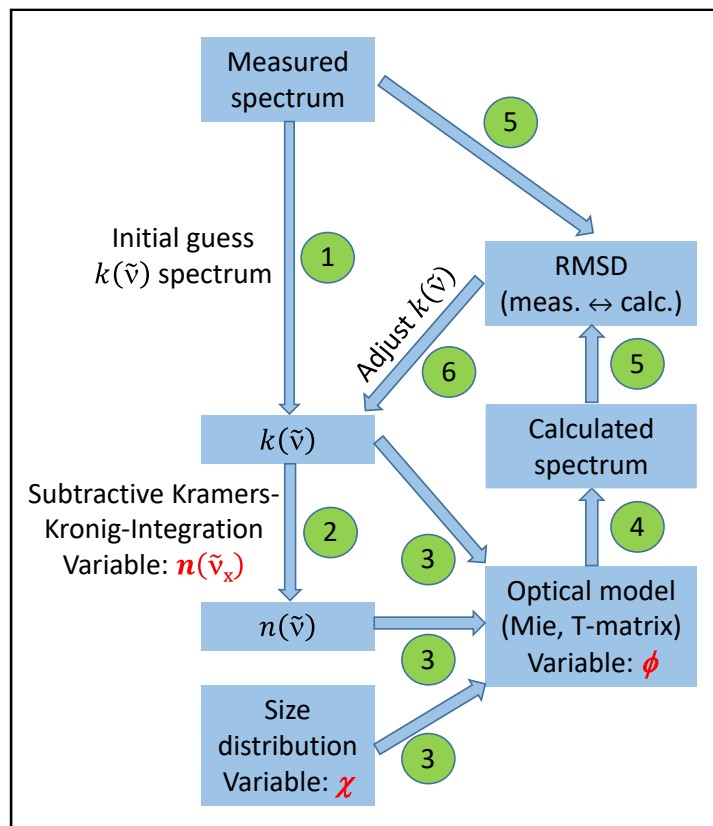

**Figure 3: Flowchart of the retrieval scheme to deduce the optical constants of the crystalline AN particles. All abbreviations, symbols, and steps in the procedure (numbered green circles) are explained in Sect. 2.2.**




**Figure 4: Real (a) und imaginary (b) parts of the complex refractive for solid AN at 223 K derived from this work (black lines) in comparison with room-temperature data from Jarzembski et al. (2003) (red lines). The insert (c) shows an expanded view of the $k$ spectrum in the regime of the $\nu_2(NO_3^-)$ mode.**




**Figure 5: Sensitivity of the retrieved $k(\tilde{v})$ spectrum on the parameters $n(\tilde{v}_x)$, $\chi$, and $\phi$ in different wavenumber regimes. (a) – (c) Variation of $n(\tilde{v}_x)$ with $\chi = 1.1$ and $\phi = 1$. (d) – (f) Variation of $\chi$ with $n(\tilde{v}_x) = 1.56$ and $\phi = 1$. (g) and (h) Variation of $\phi$ with $n(\tilde{v}_x) = 1.56$ and $\chi = 1.1$.**




**Figure 6: (a) AN mass concentrations retrieved from GLORIA measurements during the StratoClim research flight on 31 July 2017 using the $n(\tilde{\nu}_x) = 1.56$, $\chi = 1.1$, and $\phi = 1$ refractive index data set (see Fig. 3b in Höpfner et al. (2019) as a comparison). Red line: aircraft altitude. (b) Difference of mean GLORIA spectra between 04:15 – 04:21 and 04:35 – 04:43 UTC at 16.5 – 16.75 km altitude (blue) and best spectral fit with the AN optical constants (orange) in the range of the $\nu_2(NO_3^-)$ mode.**






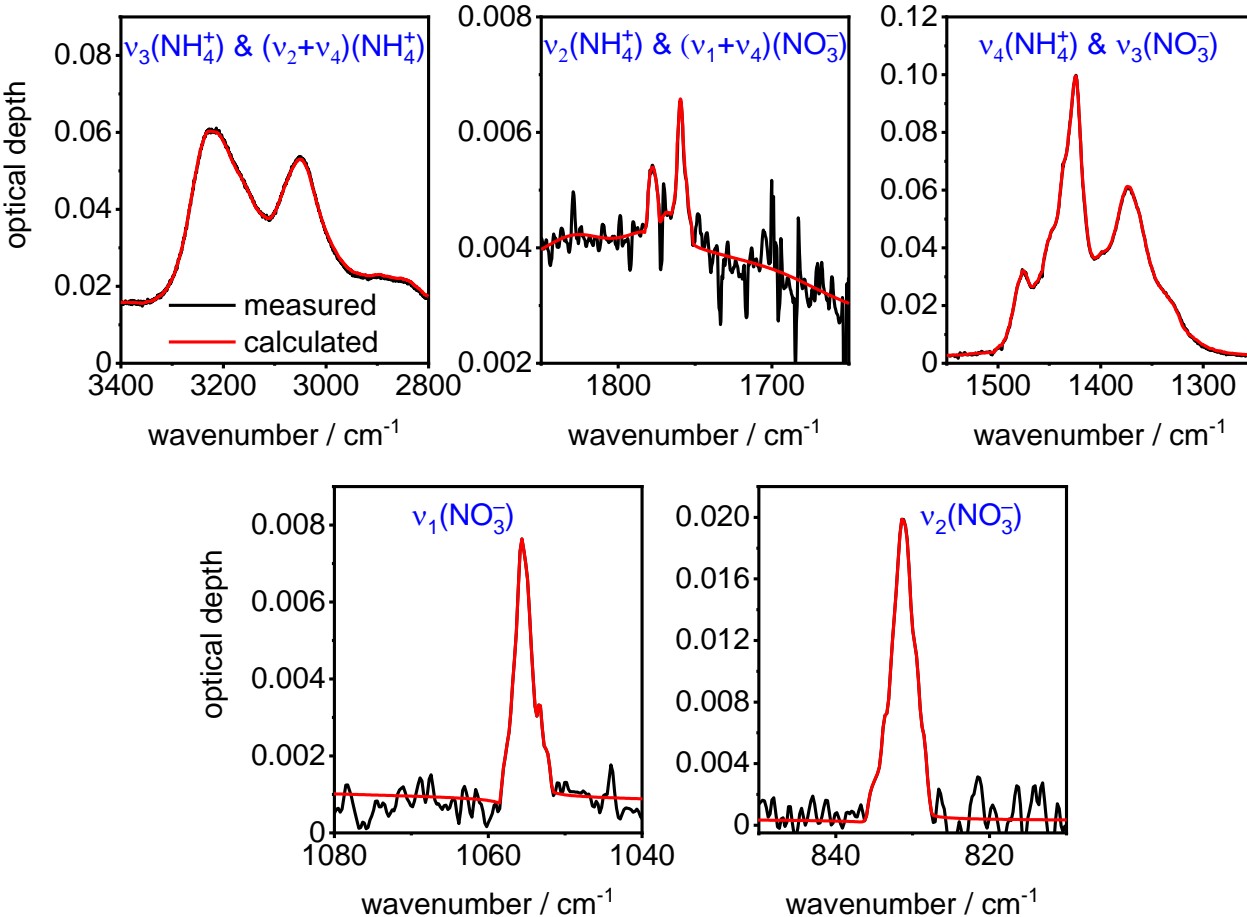

**Figure A1: Comparison between the measured (black lines) and the computed (red lines) infrared extinction spectrum of crystalline AN particles in five different wavenumber regimes after convergence of the minimization algorithm with $n(\tilde{\nu}_x) = 1.56$, $\chi = 1.1$, and $\phi = 1$. The notation of the AN vibrational modes is according to Fernandes et al. (1979).**


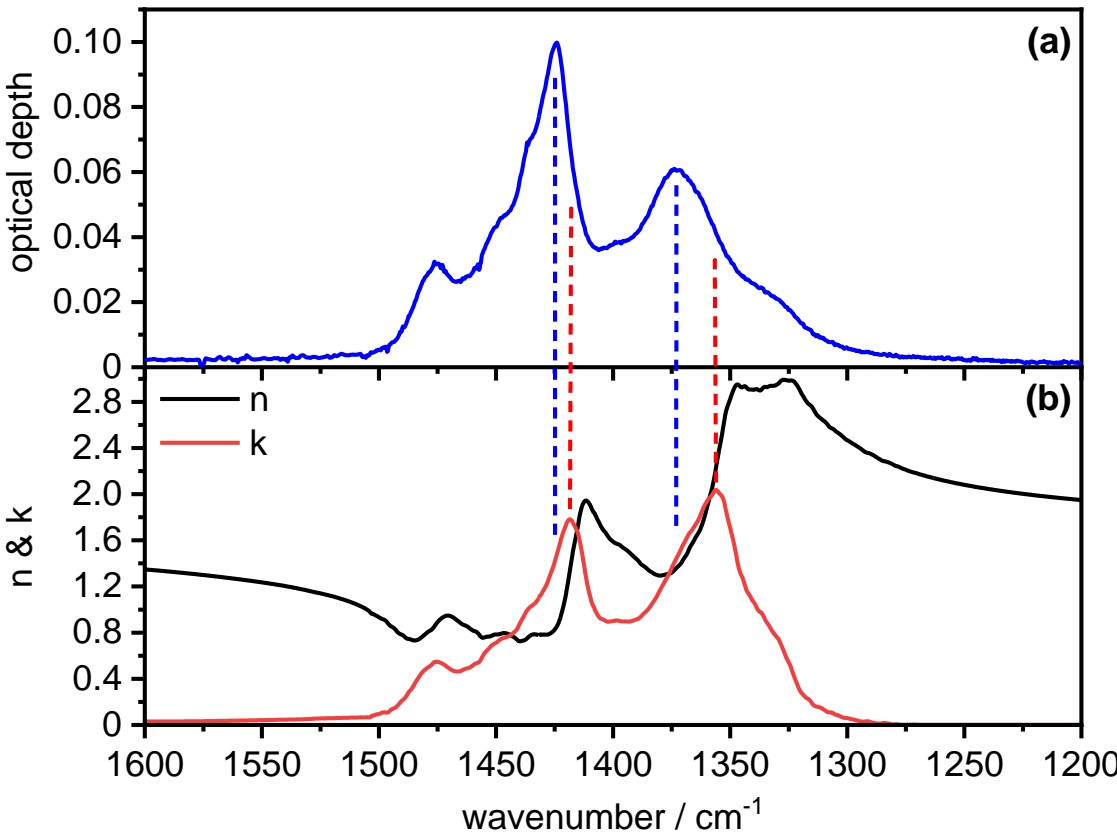

**Figure B1: Comparison between the measured infrared extinction spectrum of crystalline AN particles (a) and the retrieved optical constants (b) in the 1200 − 1600 cm⁻¹ wavenumber range. The retrieval was performed with the parameters $n(\tilde{\nu}_x) = 1.56$, $\chi = 1.1$, and $\phi = 1$. Vertical dashed lines indicate peak positions in the particle extinction spectrum (blue) and the $k(\tilde{\nu})$ spectrum (red).**






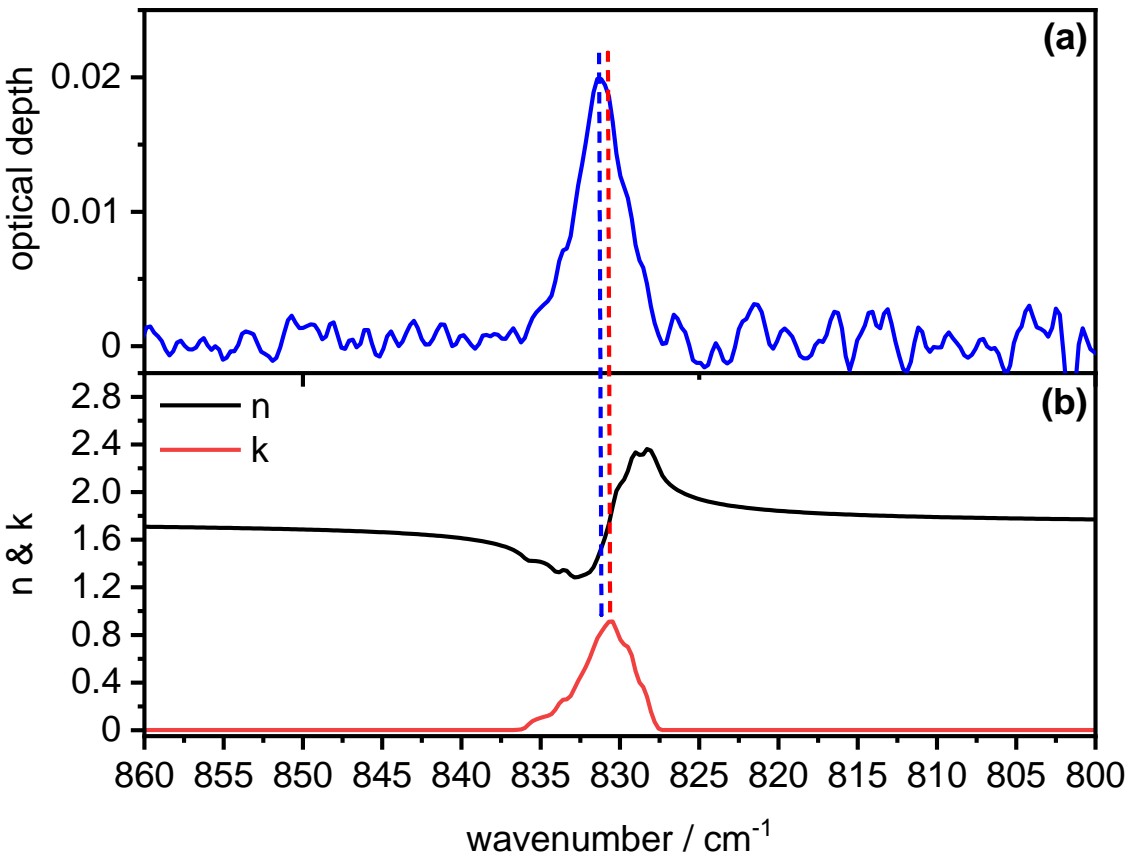

**Figure B2: Comparison between the measured infrared extinction spectrum of crystalline AN particles (a) and the retrieved optical**
**constants (b) in the 800 – 860 cm⁻¹ wavenumber range. The retrieval was performed with the parameters $n(\tilde{v}_x) = 1.56$, $\chi = 1.1$, and $\phi = 1$. Vertical dashed lines indicate the peak position in the particle extinction spectrum (blue) and the $k(\tilde{v})$ spectrum (red).**