# Peer review of "High-resolution optical constants of crystalline ammonium nitrate for infrared remote sensing of the Asian Tropopause Aerosol Layer"

_Atmospheric Measurement Techniques, 2020_

## Referee Comment (RC1) · Anonymous Referee #1 · 5 Dec 2020

Excellent experimental work. Very thorough and complete. The methodology for the retrieval of the optical constants is also satisfactory.

---

## Referee Comment (RC2) · Anonymous Referee #2 · 22 Dec 2020

Spectroscopic methods are often use to probe regions of the atmosphere that are not easy to get to, so to speak. The ability to use these methods to obtain information on the composition, number, size and shape of atmospherically important compounds relies on the availability of high-quality, wavelength-dependent complex refractive indices. This manuscript contributes nicely to the library of such data, this time for ammonium nitrate (AN) at low temperature. The authors have carried out a carefully planned experiment and have followed it up with a detailed optical analysis that sets some basic limits on the applicability of the refractive indices with respect to the value of the real index at high wavenumber as well as shape factor and aspect ratio. To no one's surprise, the optical constants at low temperature are strikingly different from those at room tem-

perature. The authors demonstrate the utility of using temperature-appropriate indices in the workup of measurements taken in the field. I for one appreciated the discussion in Appendix B regarding the shifting of peaks in the fingerprint region below 1500 cm-1.

Now for some general comments...

1.) Can the authors provide a sense of how many spectra were recorded and how long the scans were? Although I do not think this is a problem when using AIDA, I was looking for verification that the size distributions of the AN particles were stable over the data collection period.

2.) With regard to the real index at the anchor point, presumably if the density of AN was available as a function of temperature, the Lorentz-Lorenz transform could be used to to get a good estimate of n at 223 K. Nevertheless, the small range used by the authors provides a general sense of the sensitivity of these types of retrievals to the anchor point value.

3.) In Appendix A, the authors note that they extended their computation below 800 cm-1 which is typical in such work to avoid truncation errors in the finite evaluation of the Kramers-Kronig integral. How far was this extension and what was k set to?

4.) Out of interest, have the authors done a retrieval for which k in the inter-band regions was not set to 0.002 (or zero beyond 3500 cm-1), i.e., performed the retrieval with all of the baseline noise in place? If so, are the inter-band values of k warranted?

5.) Additionally, what is meant by the occurrence of "singular spikes" which required the application of a smoothing function. Did they originate in the spectrum from which the initial estimate of k was derived?

Again, I congratulate the authors on a well-done study.

---

## Author Comment (AC2) · 15 Jan 2021

**Answer to Anonymous Referee #2**

Thank you very much for your positive evaluation of our manuscript and the helpful comments. Below, we address your individual comments and describe the associated changes made in the revised manuscript version.

**Anonymous Referee #2**

Spectroscopic methods are often use to probe regions of the atmosphere that are not easy to get to, so to speak. The ability to use these methods to obtain information on the composition, number, size and shape of atmospherically important compounds relies on the availability of high-quality, wavelength-dependent complex refractive indices. This manuscript contributes nicely to the library of such data, this time for ammonium nitrate (AN) at low temperature. The authors have carried out a carefully planned experiment and have followed it up with a detailed optical analysis that sets some basic limits on the applicability of the refractive indices with respect to the value of the real index at high wavenumber as well as shape factor and aspect ratio. To no one's surprise, the optical constants at low temperature are strikingly different from those at room temperature. The authors demonstrate the utility of using temperature-appropriate indices in the workup of measurements taken in the field. I for one appreciated the discussion in Appendix B regarding the shifting of peaks in the fingerprint region below 1500 cm-1.

Now for some general comments...

1.) Can the authors provide a sense of how many spectra were recorded and how long the scans were? Although I do not think this is a problem when using AIDA, I was looking for verification that the size distributions of the AN particles were stable over the data collection period.

Yes, in the AIDA chamber the number concentration of the AN aerosol particles was very stable during the period where the size distribution measurements and FTIR scans were conducted. Due to sedimentation and sampling losses, the number concentration of the crystallized AN particles decreased by about 5% per hour. A combined size distribution measurement with the SMPS and APS instruments took about 6 min, so the number concentration of the AN particles only varied by about 0.5% within that period. For the FTIR measurements, we adjusted the number of scans such that the overall recording time was also about 6 min, overlapping with the period where the size distribution of the AN particles was measured. The spectrum shown in Fig. 2a was thus averaged over 500 scans at 0.5 cm-1 resolution. We will add this information to the revised manuscript version:

Line 114: "500 individual scans were averaged for each spectrum."

Line 140: "Note that these two measurements were carried out in an overlapping time range and lasted about 6 min. During this time,  $N_{aer}$  varied by less than 0.5%."

2.) With regard to the real index at the anchor point, presumably if the density of AN was available as a function of temperature, the Lorentz-Lorenz transform could be used to to get a good estimate of n at 223 K. Nevertheless, the small range used by the authors provides a general sense of the sensitivity of these types of retrievals to the anchor point value.

Yes, that would be a good way to better define the value for the anchor point. However, we must also consider the polymorphic phase change of the solid AN particles from phase IV to V when going to lower temperatures, which might also affect the density and the refractive index (but we did not find any measurements in the literature). Since we could not limit this value any better at this time, we thought it best to define a reasonable range of uncertainty for the anchor point value and examine its impact on the retrieval results.

3.) In Appendix A, the authors note that they extended their computation below 800 cm-1 which is typical in such work to avoid truncation errors in the finite evaluation of the Kramers-Kronig integral. How far was this extension and what was k set to?

Yes, we will extend this discussion in the revised manuscript version. As said in line 315, we used the room-temperature *k* data from Jarzembski et al. (2003) for the extension. These data extend to 500 cm-1, but only show one further, spectrally narrow absorption band at 717 cm-1 with a maximum *k* value of about 0.1, thus ten times smaller than the  $v_2(NO_3^-)$  mode of AN at 831 cm-1. The influence of the non-measured range below 800 cm-1 on the Kramers-Kronig integral was therefore rather small. For the extension, we considered the 800 – 690 cm-1 range to fully capture the additional 717 cm-1 band, interpolated the Jarzembski et al. (2003) *k* data to our wavenumber resolution, and added them to our measured spectrum below 800 cm-1 prior to performing the Kramers-Kronig integration. We will extend our statement on line 315 as follows:

"These data are available down to 500 cm-1, but just show one further, spectrally narrow absorption band at 717 cm-1 with a maximum *k* value of about 0.1. To fully capture this mode, it was sufficient to consider the range from 800 to 690 cm-1 for the extension. We interpolated the Jarzembski et al. (2003) *k* data in this range to the resolution of our measurements and added them to the  $k(\tilde{v})$  spectrum below 800 cm-1 before the Kramers-Kronig integration."

4.) Out of interest, have the authors done a retrieval for which k in the inter-band regions was not set to 0.002 (or zero beyond 3500 cm-1), i.e., performed the retrieval with all of the baseline noise in place? If so, are the inter-band values of k warranted?

We have first tried to perform the retrieval with the original data, but it has led to a considerable noise in the k data in the inter-band regions and was also extremely time-consuming because

our approach requires that each wavenumber point is individually optimized (see also answer to point 5 below). Smoothing and/or setting the inter-band values to a small residual value was therefore the only practicable solution. For the wavenumber ranges between 3500 and 2800 cm-1 and between 2000 and 800 cm-1, we have included all relevant spectral subsections with distinct AN infrared bands (as also tabulated in Fernandes et al., 1979) in the full *k* retrieval, meaning that there should not be any unconsidered absorption in the inter-band regions where we have set *k* to 0.002. However, the situation is different in the range between 2800 and 2000 cm-1, where absorption is generally very low, but some very weak infrared bands, mostly combination modes, have been assigned by Fernandes et al. (1979) (see Table 3 therein). Here, it is important to say that we do not resolve these modes with our approach, and we will do so in the revised manuscript version, line 369:

"Note that Fernandes et al. (1979) have assigned further, very weak infrared bands of AN, mostly combination modes, in the region between 2000 and 2800 cm-1. We were not able to detect these signatures in our measured spectrum, hence, these bands are not represented in our retrieved  $k(\tilde{v})$  spectrum."

5.) Additionally, what is meant by the occurrence of "singular spikes" which required the application of a smoothing function. Did they originate in the spectrum from which the initial estimate of k was derived? Again, I congratulate the authors on a well-done study.

We assume that the occurrence of these "spikes" in the *k* spectrum results from the nature of our optimization method, where we had to include each individual wavenumber grid point in the retrieval. Sometimes the routine (optimization code) got stuck (i.e., found a local minimum) when there was still considerable noise, or even "spikes" in the retrieval result for *k*. A much better and faster convergence to the "global" minimum was obtained when applying a weak smoothing function. A better formulation than "to avoid the occurrence of singular spikes in the  $k(\tilde{v})$  spectrum" (line 362) would therefore be "to improve the convergence behavior of the optimization algorithm" – we will change this in the revised manuscript version.

Ideally, as briefly outlined in Appendix A and elaborated in detail in some of the cited articles (e.g. Ossenkopf et al., 1992; Segal-Rosenhaimer et al., 2009), the retrieval of optical constants from aerosol infrared spectra would involve particles which are small enough to be treated with Rayleigh theory. In principle, smaller-sized AN crystals could easily be generated by reducing the solute concentration in the nebulizer that we used for aerosol generation. Only, the injection period to fill the huge volume of the AIDA chamber with such small particles and achieve a sufficient mass loading for a good signal-to-noise ratio in the infrared spectrum even for small absorption bands would just be too long. Hence, we had to use larger AN particles and the iterative approach the deduce the optical constants.